# Recent Advances on Iron(III) Selective Fluorescent Probes with Possible Applications in Bioimaging

**DOI:** 10.3390/molecules24183267

**Published:** 2019-09-07

**Authors:** Suban K. Sahoo, Guido Crisponi

**Affiliations:** 1Department of Applied Chemistry, S.V. National Institute Technology, Surat 395007, Gujrat, India; 2Dipartimento di Scienze Chimiche e Geologiche, Università di Cagliari, 09042 Monserrato, Italy; crisponi@unica.it

**Keywords:** Fluorescent sensors, Turn-on sensors, Ratiometric sensor, iron(III), Bioimaging

## Abstract

Iron(III) is well-known to play a vital role in a variety of metabolic processes in almost all living systems, including the human body. However, the excess or deficiency of Fe^3+^ from the normal permissible limit can cause serious health problems. Therefore, novel analytical methods are developed for the simple, direct, and cost-effective monitoring of Fe^3+^ concentration in various environmental and biological samples. Because of the high selectivity and sensitivity, fast response time, and simplicity, the fluorescent-based molecular probes have been developed extensively in the past few decades to detect Fe^3+^. This review was narrated to summarize the Fe^3+^-selective fluorescent probes that show fluorescence enhancement (turn-on) and ratiometric response. The Fe^3+^ sensing ability, mechanisms along with the analytical novelties of recently reported 77 fluorescent probes are discussed.

## 1. Introduction

Iron is well known to play an important role in a variety of physiological processes in the human body, such as oxygen metabolism, muscle contraction, synthesis of DNA and RNA, nerve conduction, proton transfer, enzyme synthesis, and regulation of acid-base balance and osmotic pressure in cells [1]. Despite the biological importance, the excess (hyperferremia) and the deficiency (hypoferremia) of iron can lead to serious health problems. The iron overload in the human body can cause severe diseases like osteoporosis, cancers, dysfunction of organs, hemochromatosis, and Alzheimer’s and Parkinson’s disease [2], whereas the iron deficiency can cause anemia and affect several cellular metabolic processes [3]. During the iron disorder, the labile iron generates destructive oxygen species (such as hydroxyl radical) via the Fenton reaction due to the facile redox process between Fe^2+^ and Fe^3+^ in the presence of molecular oxygen. The reactive oxygen species can damage peroxidative tissue and cause serious complications in pathological situations like *β*-thalassemia [4]. Therefore, there is a need for novel analytical methods for the monitoring of iron concentration in various environmental, industrial, and biological samples.

The optically (chromogenic and fluorogenic) active molecular probes have been widely investigated for the selective detection of Fe^3+^ in the last few decades. Among the two optical modes, the fluorescence-based molecular probes are extensively developed because of their simplicity, high selectivity and sensitivity, precise and real-time measuring of the target analyte up to a very low concentration without the need of pre-treatment of the sample, and sophisticated instrument [5]. The fluorescent probes are mainly designed by suitably connecting the chelating agent (binding unit) with a light-emitting group (fluorophore unit) (Scheme 1). The selective complexation of target analyte with the binding unit alters the fluorescence property of the photoexcited fluorophore mainly due to the energy or electron transfer, which allows quantifying the target analyte. The fluorescence signals from the probe upon analyte binding can be observed in the form of enhancement (turn-on), quenching (turn-off), or red/blue-shift in the fluorescence maxima of the probe (ratiometric). The well-known mechanisms like fluorescence resonance energy transfer (FRET), photo-induced electron transfer (PET), intramolecular charge transfer (ICT), C=N isomerization, chelation-induced enhanced fluorescence (CHEF), excimer formation, etc. have been applied to develop fluorescent probes for the selective detection of various analytes, including Fe^3+^, and the mechanisms are well described in the recently published review paper [6].

In 2012, we reviewed the various molecular and supramolecular fluorescent probes developed for the selective detection of Fe^3+^ [7] and observed that most of the fluorescent probes are based on the fluorescent quenching mechanism due to the paramagnetic nature of Fe^3+^ [8,9]. The fluorescent turn-on and ratiometric probes possess several analytical novelties like less probability to give false signals, increased sensitivity over turn-off probes, and, therefore, several Fe^3+^-selective fluorescent turn-on and ratiometric probes are reported in the last few years. This critical review was narrated to summarize the Fe^3+^-selective fluorescent turn-on and ratiometric probes developed after 2012, and discussion has been made on the sensing mechanisms with their potential applications to biological samples for the qualitative and quantitative monitoring of intracellular Fe^3+^ ions in live cells. All the fluorescent probes were presented in three different groups: (i) fluorescent turn-on probes for Fe(III), (ii) fluorescent ratiometric probes for Fe(III), and (iii) fluorescent chemodosimeters for Fe(III) according to their signaling process and sensing mechanisms.

## 2. Fluorescent Turn-on Probes for Fe(III)

The paramagnetic nature of the Fe^3+^ is well known to quench the fluorescence from the organic fluorophores and, therefore, the majority of the reported Fe^3+^-selective fluorescent probes show fluorescent turn-off responses. Also, it is challenging to develop fluorescent probes for an iron that showed turn-on and/or ratiometric fluorescent responses. The fluorescent probes with the turn-on response have several analytical advantages like high sensitivity, low background, and their potential applications in live-cell imaging. The literature survey revealed that the rhodamine/fluorescein derivatives are extensively applied for the development of fluorescent turn-on probes for various ionic and neutral analytes because of the reversible fluorescence changes with respect to the structural changes occurred in the spirocyclic ring [10]. The closed ring spirocyclic form of the rhodamine is colorless and non-fluorescent, while the open-ring spirocyclic form is highly fluorescent and colored. The general mechanism to design rhodamine/fluorescein-based fluorescent turn-on probes for Fe^3+^ is described in Scheme 2. The carboxyl frame of rhodamine is converted into a closed ring form by reacting with small molecules possessing –NH_2_ group, and then the complexation-induced opening of the spirocyclic ring results in strong fluorescence emission. With this mechanism, the recently developed rhodamine-based Fe^3+^-selective fluorescent probes **1**–**40** are summarized in Table 1.

The summarized rhodamine-based probes in Table 1 detect Fe^3+^ either in aqueous or semi-aqueous medium, as well as within live cells by forming complex either in 1:1 or 1:2 ratio followed by the opening of the spirocyclic ring of the probes. The mechanism of sensing for all the probes is similar, as described in Scheme 2. The reversible fluorescent probe 1 forms complex with Fe^3+^ in 1:1 binding stoichiometry and opens the spirocyclic ring (Scheme 3), which allows to detect Fe^3+^ down to 0.1 µM [11]. This probe shows stable fluorescence over pH 3.5–8.2. The probe shows promising results to locate the intracellular Fe^3+^ ions in live SH-SY5Y cells in real-time. Also, the probe has been applied to detect the labile Fe^3+^ pools in mitochondria and endosomes/lysosomes of SH-SY5Y cells (Figure 1). The probe 2 has been applied to detect Fe^3+^ down to 50 nM and non-cytotoxic up to 6 μM [12]. Because of its high specificity from amino acids, BSA protein, and human blood serum, the probe **2** is useful in monitoring intracellular Fe^3+^ ions concentration. The probe **3** with a quinoline moiety bound to rhodamine 6G hydrazide shows good cell permeability and the ability to locate the subcellular distribution of Fe^3+^ in EJ (lung cancer) cells by fluorescence imaging experiments [13]. The probe **4** detects Fe^3+^ concentration down to 2.2 µM and is suitable between the pH 6–7.5 [14]. This cell-permeable probe has been applied to image intracellular Fe^3+^ ions in HeLa cells. The probe **5** shows permeability of the plasma membrane to rhodenal and potential to locate the iron pools in the cells [15]. The probe **6** can detect Fe^3+^ over wide pH ranges from 5 to 11 with the minimum detection limit estimated down to 3 µM [16]. Probe **6** has been applied for bioimaging experiments in L-929 cells (mouse fibroblast cells) and BHK-21 (hamster kidney fibroblast), revealing good biocompatibility, cell permeability, and minimum toxicity.

The probe **7** selectively forms a complex with Fe^3+^ in 1:1 stoichiometry and opens the spirocyclic ring to give significant fluorescence turn-on response [17]. Probe **7** can detect Fe^3+^ down to 0.031 µM, and the in situ generated **7**-Fe^3+^ complex has been applied for the selective sensing of S^2-^ anions. The probe **8** is useful in detecting Fe^3+^ in the biologically relevant pH from 6 to 9 and has shown very low cytotoxicity [18]. A confocal fluorescence imaging study of **8** reveals good cell permeability and the ability to monitor intracellular Fe^3+^ in live cells. The probes **9** and **10** show similar high selectivity towards Fe^3+^ with the detection limit down to 66 nM and 44.5 nM, respectively [19]. The probe **11** bearing the di-2-picolylamine as a binding unit shows high selectivity towards Fe^3+^, eliminates the Cr^3+^ interference during Fe^3+^ detection, and the selectivity is maintained over the pH range 6 to 7.5 [20]. The Fe^3+^-selective probe **12** shows good linear fluorescence response from 2 µM to 20 µM with the limit of detection (LOD) estimated down to 0.32 µM [21]. The probes **13** and **14** show linearity range from 0.9–20 µM and 5–20 µM with the detection limit down to 0.9 µM and 5 µM, respectively [22]. The probe **15** alone is not-fluorescent above pH = 6, but the selective complexation with Fe^3+^ opens the spirocyclic ring to give significant turn-on fluorescence at 581 nm [23]. The probe **15** can detect Fe^3+^ down to 0.396 μM and is applied successfully for detecting Fe^3+^ in human liver cells (L-02) and rat neuronal (PC12) cells.

The rhodamine-triazine aminopyridine derivative **16** shows the detection limit of 41 nM for Fe^3+^, and the probe has been applied to monitor Fe^3+^ ions in real water samples and living HL-7702 cells [24]. The probe **17** is applied for the cascade detection of Fe^3+^ and the thiols (glutathione, homocysteine, cysteine) in solution and live cells [25]. The reversible fluorescent probe **18** shows turn-on fluorescence response between 10 and 70 μM of Fe^3+^ with the detection limit of 0.195 ppm [26]. The probe **19** is suitable to detect Fe^3+^ in the pH range from 4 to 7, with the estimated LOD of 0.26 μM [27]. The fluorescent enantiomer **20** shows a detection limit of 183 nM Fe^3+^ and detects Fe^3+^ in living cells with low cytotoxicity [28]. The probe **21** is ideal for detecting Fe^3+^ in the pH range from 4 to 8, with the LOD of 57 nM [29]. Also, the fluorescence turn-on response from the **21**-Fe^3+^ complex formed in solution is shown to reverse upon addition of Na_4_P_2_O_7_. The benzothiazole-functionalized fluorescein derivative **22** shows nanomolar detection limit (7.4 nM) for Fe^3+^ with the potential to detect intracellular Fe^3+^ ions in live Hep G2 cells with low cytotoxicity [30].

The thiophene-modified rhodamine 6G derivative **23** shows selectivity towards Fe^3+^ and Al^3+^ with the LOD of 5 and 6 μM [31]. The rhodamine-furan-5-carbaldehyde chemosensor **24** shows high Fe^3+^ selectivity with the LOD of 17 nM. The probe **24** is safe for biological use and is non-toxic to living cells [32]. The turn-on colorimetric and fluorescent sensor **25** shows the LOD 0.768 μM Fe^3+^ and bioimage Fe^3+^ ions in live HeLa cells [33]. The probe **26** detects Fe^3+^, Al^3+^, and Cr^3+^ with the estimated LOD of 0.29, 0.34, and 0.31 μM, respectively [34]. The probe **26** has been applied to mimic the Boolean logic gates with two and four inputs. The probe **26** has been successfully applied to monitor the selective cations and also the native cellular iron pools. The fluorescence of rhodamine-2-thioxoquinazolin-4-one derivative **27** is increased linearly at 555 nm with the addition of Fe^3+^ from 0 to 75 μM, and the LOD is estimated down to 4.11 μM [35]. The concentration of Fe^3+^ is determined in various real water samples and is successfully applied to monitor intracellular Fe^3+^ ion in living cells. The probe **28** is designed by reaction rhodamine hydrazide with two equivalents of 2-(thiophen-2-yl)acetyl chloride [36]. The turn-on fluorescence from **28** can be applied to detect Fe^3+^ down to 0.13 μM, and the probe is suitable in the pH range from 4 to 9. The probe **29** shows a good linearity range from 0.8 to 20 μM with the estimated LOD of 11.6 nM [37]. The probe **30,** possessing rhodamine and anthracene, detects Fe^3+^ down to 42 nM and has been applied successfully to monitor Fe^3+^ ions in living cells and zebrafish (Figure 2) [38]. The furfuran-based rhodamine B fluorescent probe 31 can be applied to detect Fe^3+^ down to 0.025 μM, and the turn-on fluorescence due to the 31-Fe^3+^ complex formation is shown to be reversed upon addition of B_4_O_7_^2^^−^ [39].

The Fe^3+^-selective fluorescent turn-on probe **32** is suitable in the pH range from 5 to 9 [40]. The probe **32** has been applied to detect basal level Fe^3+^ and the dynamic changes in Fe^3+^ levels in live bovine aortic endothelial cells (BAEC) at a subcellular resolution that reveal two Fe^3+^ pools in endosomes/lysosomes and mitochondria. The probe **33** forms complex with Fe^3+^ in 2:1 binding ratio and shows significant fluorescence enhancement at 588 nm [41]. The probes detect Fe^3+^ better in the basic condition (pH 7 to 10), and the estimated LOD is 92 nM. The probe **34** fluorescence enhances linearly from 1 to 170 μM of Fe^3+^ with the estimated LOD of 1.2 μM [42]. The rhodamine-based quinoline conjugated probe **35** shows distinct UV-Vis spectral changes upon addition of Fe^3+^ and Cu^2+^, but the fluorescence is enhanced selectively in the presence of Fe^3+^ [43]. The turn-on fluorescence from **35** allows to detect Fe^3+^ down to 33 nM, and the probe has been applied to detect Fe^3+^ ions in zebrafish embryos. The furan-2-carbonyl chloride modified rhodamine B derivative **36** selectively forms a complex with Fe^3+^ in 1:1 ratio and shows significant fluorescence enhancement at 582 nm that allows to detect Fe^3+^ down to 0.437 μM [44]. The pyridine-type rhodamine B fluorescent probes **37** and **38** are developed to detect Fe^3+^ down to 0.067 μM and 0.345 μM, respectively [45]. Its analytical applicability has been tested by monitoring Fe^3+^ concentrations in various real water samples and live cells. The probe **39**, developed by combining rhodamine and piperonaldehyde, shows LOD of 11.8 nM Fe^3+^ [46]. The rhodamine-based probe **40** shows high selectivity towards Fe^3+^ (LOD = 0.205 μM) and detects the intracellular Fe^3+^ ions in HeLa cells [47].

The mechanisms like CHEF, PET, excimer formation, C=N isomerization, ESIPT (excited-state intramolecular proton transfer), ICT, etc. are also applied to develop Fe^3+^-selective fluorescent turn-on probes (Table 2). Belfield and his co-workers [48] introduced a novel PET-based reversible fluorescence turn-on probe **41** for the selective detection of Fe^3+^. The probe **41** was designed by connecting boron-dipyrromethene (BODIPY) fluorophore with a 1,10-diaza-18-crown-6-based cryptand that acts as the analyte binding unit. The weakly fluorescent **41** (PET process is active) showed significant fluorescence enhancement at 512 nm (λ_exc_ = 480 nm) upon addition of Fe^3+^ in H_2_O-CH_3_CN (9:1 *v*/*v*). The fluorescence enhancement occurred due to the inhibition of the PET from cryptand to BODIPY fluorophore upon complexation with Fe^3+^. The probe showed a LOD of 1.3 × 10^−7^ M and was applied for the detection of intracellular Fe^3+^ ions in living HCT-116 cells. Using the Calix [4] arene framework, the quinoline-appended dipodal fluorescent probe **42** has been developed, and its cations sensing ability has been examined in CH_3_CN [49]. Upon complexation of Fe^3+^ with **42** in 1:1 binding ratio, significant fluorescence enhancement has been observed at 418 nm (λ_exc_ = 310 nm), and the LOD of 0.334 μM Fe^3+^ has been estimated from the fluorescence titration experiment. Further, the **42**-Fe^3+^ complex has been applied as an intracellular fluorescent agent in MDA-MB-231 cells.

Nandre et. al. [50] developed a novel fluorescent probe **43** based on the benzo-thiazolo-pyrimidine unit for the selective turn-on sensing of Fe^3+^ in aqueous acetonitrile medium. The probe **43** showed a remarkable fluorescence enhancement at 554 nm (*λ*_exc_ = 314 nm) in the presence of Fe^3+^ due to the inhibition of PET. The sensor formed a host-guest complex in 1:1 stoichiometry with the limit of detection down to 0.74 nM. Further, the sensor was successfully utilized for the qualitative and quantitative intracellular detection of Fe^3+^ in live HepG2 cells and HL-7701 cells by a confocal imaging technique (Figure 3). The diketopyrrolopyrrole-based supramolecular fluorescent probe **44** shows selective response in the presence of Fe^3+^ and Au^3+^ [51]. In EtOH/0.01 M PBS buffer (*v*/*v*, 1:1, pH 7.4), the weakly emission from **44** shows significant enhancement at 578 nm (465 nm) due to the inhibition of C=N isomerization at the excited state due to the formation of a **44**-Fe^3+^ complex in 1:2 ratio. From the fluorescence enhancement, the LOD for Fe^3+^ has been estimated as 8 nM, and the probe is suitable to detect Fe^3+^ over a pH range from 3 to 8. Besides, the probe **44** is cell-permeable and detects the intracellular Fe^3+^ concentration in human lung adenocarcinoma cells (A549).

The macrocyclic-based fluorescent probe **45** containing anthracene fluorophore shows weak emission at 398, 421, and 447 nm (λ_exc_ = 373 nm) in Tris-HCl buffer (20 mM, pH 7.2) containing 50% methanol (*v*/*v*) [52]. In the presence of Fe^3+^, the formation of a **45**-Fe^3+^ complex in 1:2 ratio inhibits the PET process, causing significant fluorescence enhancement. Probe **45** shows a linear response range from 1 μM to 10 μM with the LOD of 0.58 μM Fe^3+^. Confocal imaging discloses that the probe **45** possesses the ability of cell membrane permeability and also the cytosolic Fe^3+^ imaging ability in SKOV-3 cells. Recently, Kim and his co-workers [53] introduced a simple anthracene-based fluorescent turn-on probe **46** substituted with 9,10-diethanolamine for the detection of Fe^3+^. In CH_3_CN:H_2_O (3:7, v/v) at pH 7, the weakly emissive probe showed emissions enhancement at 406, 429, and 456 nm characteristic of the anthracene monomer (λ_exc_ = 376 nm). Experimental results revealed that the probe **46** formed a complex with Fe^3+^ in 1:2 binding stoichiometry with the association constant of 9.29 × 10^6^ M^−1^. The chelation-induced enhanced fluorescence (CHEF) effect along with the inhibition of PET resulted in the fluorescence enhancement. The limit of detection of **46** for Fe^3+^ was estimated down to 0.1 pM, and the probe was applied for the monitoring of Fe^3+^ ions in *Candida albicans* (C.A., KCTC-11282) cells. Further, the **46**-Fe^3+^ complex ensemble was applied for the selective detection of CN^-^, and also an INHIBIT type logic gate was proposed by taking the two inputs, Fe^3+^ and CN^-^.

The novel quinolone-based fluorescent turn-on probe **47** has been developed for the detection of Fe^3+^ and Cr^3+^ [54]. The analytical study of **47** towards Fe^3+^ exhibits a significant fluorescence enhancement at 458 nm (λ_exc_ = 420 nm). The formation of a **47**-Fe^3+^ complex in 2:1 ratio restricts the rotation of thiophene, resulting in the fluorescence enhancement both in ethanol and aqueous medium. The probe shows LOD of 1 μM for Fe^3+^. The probe has been applied for the biological applications in live HepG2 cells to monitor intracellular Fe^3+^, and also the probe has been applied to detect the autophagosome-lysosome fusion during the autophagy process. The restriction of molecular rotation after forming aggregation also results in significant fluorescent enhancement. The pyrene-based Schiff base **48** solution in CH_3_CN undergoes nano-aggregation by adding poor solvent water and shows significant emission enhancement at 465 nm due to the aggregation-induced emission enhancement (AIEE) [55]. The cations sensing ability of **48** in CH_3_CN has been tested by adding different metal ions from their water solution, revealing significant fluorescence enhancement at 500 nm (λ_exc_ = 395 nm) in the presence of Fe^3+^, Cr^3+^, and Al^3+^. The fluorescence enhancement in the presence of the selective trivalent metal ions has occurred due to the formation of pyrene excimer upon complexation between **48** and Fe^3+^/Cr^3+^/Al^3+^ in 2:1 ratio. With **48**, the LOD is estimated down to 0.106 μM, 0.111 μM, and 0.117 μM for Fe^3+^, Cr^3+^, and Al^3+^, respectively. Besides, the probe shows the ability to detect the selective metal ions within the live Raw264.7 cells. With the excimer formation mechanism, the pyrene-based fluorescent probe **49** has been developed for the selective detection of Fe^3+^ [56]. In acetonitrile-acetone (*v*/*v* = 99:1), the probe **49** shows weak emission due to the transfer of the electrons on the nitrogen atom to pyrene (PET active). However, upon complexation with Fe^3+^ in 1:1 binding ratio, the probe shows fluorescence enhancement at 507 nm due to the formation of pyrene excimer (λ_exc_ = 382 nm). The quantum yield of the probe (*Φ* = 0.001) is enhanced 41-fold (*Φ* = 0.041) upon complexation. Besides, the Fe^3+^-directed formation of a pyrene excimer has also been detected in live HeLa cells.



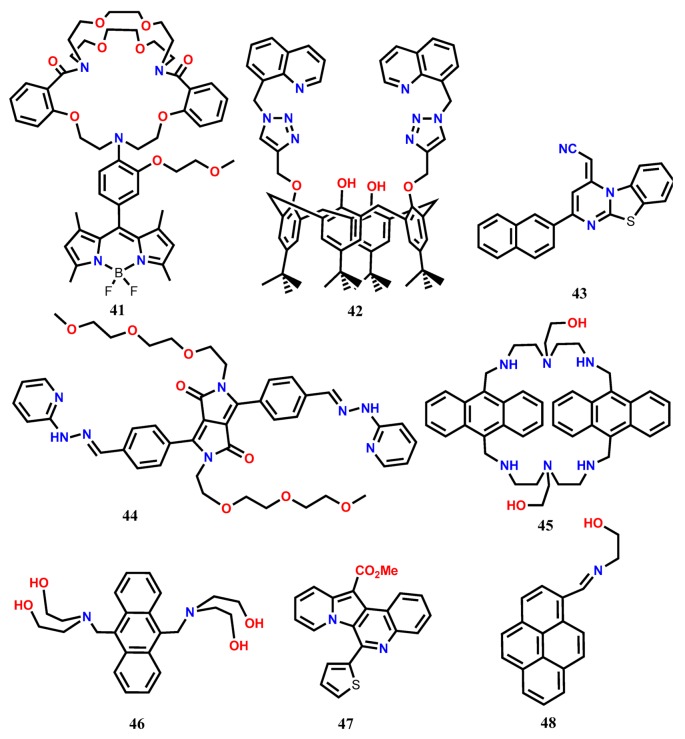



Han et al. [57] introduced a novel naphthalimide-diethylenetriamine-quinoline-based fluorescent turn-on probe **50** for the selective detection of Fe^3+^ that operated with AIEE and PET mechanisms. The weakly emissive **50** at 513 nm (λ_exc_ = 403 nm) in pure CH_3_CN showed maximum fluorescence enhancement along with the red-shift from 513 to 524 nm in CH_3_CN containing 70% water due to the formation of nano-aggregates, and the red-shift is due to the restriction of intramolecular rotation. The fluorescent organic nanoparticles (FONs) of **50** showed stable fluorescence in the pH interval from 7–14 and also showed selective fluorescence enhancement in the presence of Fe^3+^. The FONs showed a linear range from 1 nM to 100 mM with the LOD of 0.35 nM Fe^3+^. It was proposed that the FONs fluorescence was disrupted due to the PET from the diethylenetriamine unit to the electron-deficient naphthalimide group. Upon complexation of **50** with Fe^3+^ in the FONs, the PET was forbidden, and a dramatic increase in fluorescent intensity was observed. In the cellular medium, the FONs showed low cytotoxicity, and the intracellular Fe^3+^ ions were detected in HeLa by using a fluorescence microscope (Figure 4).

Dwivedi et al. [58] utilized the naphthalimide as a signaling unit and the suitably connected thiophene and piperazine rings as recognition unit to develop a highly selective fluorescent turn-on probe **51** to detect Fe^3+^ in solution and live cells. The PET active probe **51** was non-fluorescent in 40% aqueous THF solution, but significant fluorescent was observed at 528 nm (λ_exc_ = 408 nm) with the addition of Fe^3+^. The S atom of thiophene unit and the N atoms of piperazine of **51** formed a complex with Fe^3+^ in 1:1 ratio, inhibiting the PET and resulting in the fluorescence enhancement. With this probe, the LOD was estimated as 0.373 μM, and the probe was applied over a wide pH range (pH 6 to 14) to detect Fe^3+^. The probe showed excellent biocompatibility and cell permeability to detect the intracellular Fe^3+^ ions in live MCF-7 cells. Further, the in-situ generated Fe^3+^-**51** complex was applied for the fluorescent turn-off sensing of AcO^−^. Applying the PET mechanism, recently, an easy-to-prepare amide-quinoline-based fluorescent probe **52** has been developed for the detection of Fe^3+^ and Al^3+^ in aqueous medium [59]. The probe forms complex with Fe^3+^ and Al^3+^ in the 1:1 binding ratio that diminishes the electron donation from the isoquinoline nitrogen atom towards the pyridyl ring and inhibits the PET. The suppression of PET at the excited state results in the significant fluorescence enhancement at 430 nm (λ_exc_ = 332 nm). The probe shows micromolar LOD of 0.092 and 0.235 μM for Al^3+^ and Fe^3+^, respectively. Also, the low cytotoxicity of **52** allows monitoring intracellular Fe^3+^ and Al^3+^ ions in live HeLa cells by fluorescence microscopy.

The Schiff-based probe **53** shows fluorescence quenching due to the excited-state intramolecular proton transfer (ESIPT) in H_2_O:EtOH (6:4, *v*/*v*) [60]. With the addition of Fe^3+^, the formation of a **53**-Fe^3+^ complex in 1:1 ratio suppresses the ESIPT and results in a significant fluorescent enhancement at 550 nm (λ_exc_ = 397 nm). The probe shows a wide linear range from 0–200 µM with the LOD of 0.8 ppb. Finally, the probe has been applied to detect intracellular Fe^3+^ ions in the cancer HeLa cells. Another Schiff base probe **54** has been developed for the selective fluorescent turn-on sensing of Fe^3+^ in CH_3_CN:H_2_O (1:1, *v*/*v*) [61]. The probe shows two weak emissions at 430 nm and 574 nm (λ_exc_ = 380 nm). The formation of a **54**-Fe^3+^ complex in 1:1 ratio facilitates the charge transfer from the imino group of **54** to Fe^3+^ ion, resulting in an enhanced emission peak centered at 482 nm. The probe can be applied to detect Fe^3+^ down to the nanomolar level (0.89 nM) and is effective at a pH range of 6 to 7. The probe shows good cells permeability, and its ability to detect intracellular Fe^3+^ ions has been tested in live HeLa cells.

Erdemir et. al. [62] introduced an anthracene-based fluorescent probe **55** containing benzothiazole group as a binding unit for the fluorescent turn-on sensing of Fe^3+^ and Cr^3+^. In CH_3_CN, probe **55** shows a weak emission at 428 nm due to the PET process (λ_exc_ = 380 nm). Addition of Fe^3+^/Cr^3+^ suppresses the PET, and the complexation of the cations with **55** in 1:2 ratio results in the formation of the anthracenyl static excimer that give distinct fluorescence at 576 nm. The fluorescence titration experiments have estimated the LOD for Cr^3+^ and Fe^3+^ as 0.46 and 0.45 µM, respectively. Finally, the probe has been applied for the fluorescence imaging of living cells and for monitoring Fe^3+^ ions in live PC-3 cells.



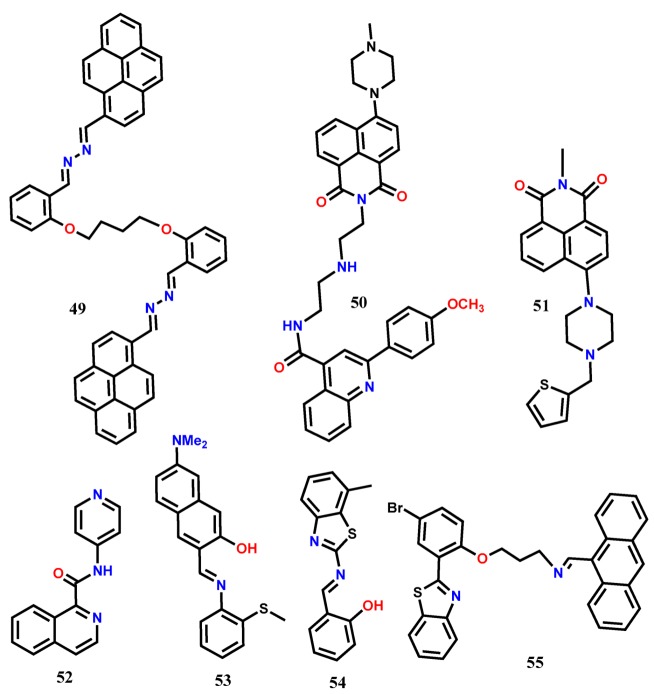



## 3. Ratiometric Fluorescent Probes for Fe(III)

The ratiometric fluorescent probes generally refer to the chemosensors that detect the target analyte from the changes in fluorescence intensity occurring at two emission bands [63]. The ratio of the change in fluorescence intensity of the probe at two different emission peaks is calibrated to monitor the target analyte. Because of the presence of two different emission bands for detection, the ratiometric probes provide several analytical advantages over the probe operated only at a single emission band and also minimize the interferences of the external environments like the concentration of the probe, instrumental parameters, photobleaching, etc. Therefore, great efforts have been given to develop ratiometric fluorescent probes with their potential applications in the field of environmental detection and biological analysis. The well-established mechanisms like fluorescence resonance energy transfer (FRET), through-bond energy transfer (TBET), excimer/exciplex formation, intramolecular charge transfer (ICT), etc. can be adopted for the designing of ratiometric fluorescent probes. Several Fe^3+^-selective ratiometric fluorescent probes have been reported in the last few years (Table 3), which are mainly based on the FRET, TBET, and ICT mechanisms.

The FRET-based probe consists of two fluorophore units separated by a spacer, where the excited state of one fluorophore (donor) transfers its energy to the closely located another fluorophore (acceptor) in a non-radiative manner, and then the energy is released in a radiative manner from the second fluorophore [63]. In designing FRET-based probes, care must be taken that the donor to acceptor distance and orientation is appropriate for energy transfer. Besides, there must be spectral overlap between the fluorescence profile of the donor fluorophore with the UV-Vis absorption spectra of acceptor fluorophore. By utilizing the napthalimide and rhodamine dyes, the FRET-based fluorescent probe **56** has been designed for the ratiometric detection of Fe^3+^ in aqueous acetonitrile (1:1, *v*/*v*, 0.01 M Tris HCl-CH_3_CN, pH 7.4) medium [64]. In this probe, the triazole appended quinoline-rhodamine conjugate acts as a selective ionophore for Fe^3+^ and FRET energy acceptor, whereas the 8-piperazinonaphthalimide moiety acts as the FRET energy donor. In the presence of Fe^3+^, the emission at 532 nm from the naphthalimide moiety is decreased, and a new emission band appears at 580 nm (λ_exc_ = 420 nm). The ratiometric response from the probe **56** is due to the complexation-induced opening of the spirocyclic ring of the rhodamine moiety that facilitates the FRET from the naphthalimide donor. The FRET is possible because of the excellent spectral overlap of the naphthalimide emission spectrum with the absorption spectrum of the rhodamine unit of **56**. With this probe, the LOD has been estimated down to 5 × 10^−8^ M (~3 ppb) and applied successfully for the monitoring of trace levels of intracellular Fe^3+^ ions in NIH 3T3 cells (Figure 5). Subsequently, the same group uses the napthalimide-rhodamine combination to develop a FRET-based multi-analytes (Fe^3+^, Al^3+^, and Cr^3+^)-selective fluorescent probe **57** [65]. This probe shows a decrease in the naphthalimide emission at 532 nm and concomitant appearance of a new emission peak at 583 nm upon addition of Fe^3+^, Al^3+^, and Cr^3+^ ions in aqueous acetonitrile (1:1, *v*/*v*, 0.01 M Tris HCl-CH_3_CN, pH 7.4) medium. The formation of the metal complex between **57** and the metal ions in 1:1 binding stoichiometry opens the spirocyclic ring of the rhodamine unit, allowing the FRET process to give the ratiometric signal. Also, the probe has been applied to detect the intracellular Fe^3+^ ions in W138 cells.

Das and his co-workers [66] developed the FRET-based fluorescent probe **58** containing a 2-hydroxynaphthalene unit as a donor and rhodamine B as an acceptor for the selective detection of Cr^3+^ and Fe^3+^ in HEPES buffered (0.1 M) EtOH-H_2_O (2:1, *v*/*v*, pH 7). The fluorescence of **58** at 455 nm (λ_exc_ = 330 nm, blue fluorescence) from the 2-hydroxynaphthalene moiety is quenched, and a new fluorescence band appears at 585 nm (red fluorescence) in the presence of Fe^3+^ and Cr^3+^. The complexation-induced ring-opening of the spirolactam unit has resulted in energy transfer from the donor to the acceptor unit. With **58**, the concentration of Cr^3+^ and Fe^3+^ can be detected down to 10 nM and 0.54 μM, respectively. Further, the probe has been applied for the detection of intracellular Cr^3+^ and Fe^3+^ ions in *Bacillus sp.* cells and *Candida albicans* cells by recording the fluorescence images. Applying the FRET mechanism, the rhodamine spirolactam has been connected to the blue fluorescent water-soluble ionic conjugated polymers (CPs) to develop a ratiometric probe for Fe^3+^ [67]. Exciting the probe **59** at 400 nm in a buffer solution (Tris-HCl, pH = 7.2), the fluorescence of CPs at 442 nm is quenched, and a new peak appears at 538 nm in the presence of Fe^3+^ due to the complexation-induced spirocyclic ring-opening of rhodamine 6G. Also, the quenching of CPs emission is due to the possible FRET to the rhodamine 6G unit. Using the ratiometric signal changes (I_538_/I_442_), the detection limit has been estimated as 0.3 μM. The FRET efficiency between CPs and rhodamine 6G is 61.8%, and the distance between the acceptor and donor as 4.06 nm, supporting the efficient FRET between the two fluorophores. Using the probe **59**, the confocal fluorescence imaging experiment has been carried to monitor the intracellular Fe^3+^ ions in live HeLa cells.

Chen et al. [68] introduced a new ratiometric fluorescent sensor **60** by suitably combining the naphthalene and rhodamine dyes for the selective detection of mitochondrial Fe^3+^ in live HeLa cells. The fluorescence of **60** at 431 nm was quenched, and concomitantly a new emission appeared at 594 nm (λ_exc_ = 371) upon addition of Fe^3+^ in EtOH-H_2_O (4:1). The ratiometric emission from the probe **60** was observed due to the possible FRET from the conjugated naphthalene donor to the rhodamine acceptor. Without any noticeable interference from other tested metal ions, the concentration of Fe^3+^ could be detected down to 6.93 μM. Besides, due to the presence of a lipophilic alkyltriphenylphosphonium (alkylTPP) cation that helps in passing directly through the phospholipid bilayers and accumulate selectively within the mitochondria inside cells, the probe **60** showed satisfactory cell permeability and detected the mitochondrial Fe^3+^ in live HeLa cells (Figure 6).

The FRET-based fluorescent probe **61** containing a dansyl unit as a donor and rhodamine 101 as an acceptor has been developed to detect Fe^3+^ in CH_3_CN-Tris buffer (9:1, *v*/*v*, pH 7.05) [69]. Fe^3+^-induced ring-opening of the spirolactam rhodamine moiety results in the formation of fluorescent derivative that can serve as the FRET acceptor (λ_exc_ = 380 nm). Ratiometric sensing of Fe^3+^ is accomplished by plotting the fluorescence intensity ratio at 605 nm and 515 nm versus Fe^3+^ ions concentration. The large Stokes shift (225 nm) shown by the probe can eliminate the back-scattering effects of excitation light. The probe displays a linear response to Fe^3+^ in the range of 5.5–25 μM with a detection limit of 0.64 μM. Combining naphthalimide and rhodamine B by using the ethylenediamine connector, a new Fe^3+^-selective FRET-based ratiometric fluorescent probe **62** has been developed [70], where the naphthalimide acts as an energy donor, while the spirocyclic ring-open form of rhodamine as the energy acceptor. Upon complexation with Fe^3+^, the emission from the naphthalimide unit of the probe **62** at 520 nm is decreased, and a significant enhancement of the characteristic fluorescence of the rhodamine is observed at 577 nm (λ_exc_ = 420 nm) in ethanol solution. The probe shows a linear range for Fe^3+^ from 0.2 to 1 μM with the LOD of 0.418 μM. Besides, the probe shows the cell permeability to detect the intracellular Fe^3+^ ions in live EC109 cells in the fluorescence imaging study.

The FRET-based benzothiazole conjugated quinoline derivative appended with rhodamine-6G ratiometric fluorescent probe **63** has been introduced for the detection of Fe^3+^ [71]. The probe shows a strong emission at 470 (λ_exc_ = 370 nm) from the benzothiazole moiety in CH_3_OH/H_2_O (2/3, *v*/*v*, pH = 7.2). The complexation-induced opening of the spirocyclic ring of the rhodamine unit results in FRET from the energy donor (benzothiazole moiety) to the energy acceptor (rhodamine-6G domain). As a result, a new peak appears at 558 nm with the gradual decrease in the intensity at 470 nm. The ratio of the emission intensities at the two wavelengths (I_558_/I_470_) exhibits good linearity with the added concentration of Fe^3+^ from 0–14 μM with the estimated LOD of 53.9 nM. The FRET-based probe **64** has been designed by suitably connecting the coumarin (energy donor) with the rhodamine moiety (energy acceptor) [72]. In EtOH-H_2_O (9:1, *v*/*v*, Tris-HCl, pH = 7.4), the receptor shows only the coumarin emission band at 475 nm (λ_exc_ = 450 nm), whereas a new emission band appears at 550 nm upon addition of Fe^3+^ due to the complexation-induced opening of the spirocyclic ring of the rhodamine. With this probe, the concentration of Fe^3+^ can be detected down to 4.05 μM.



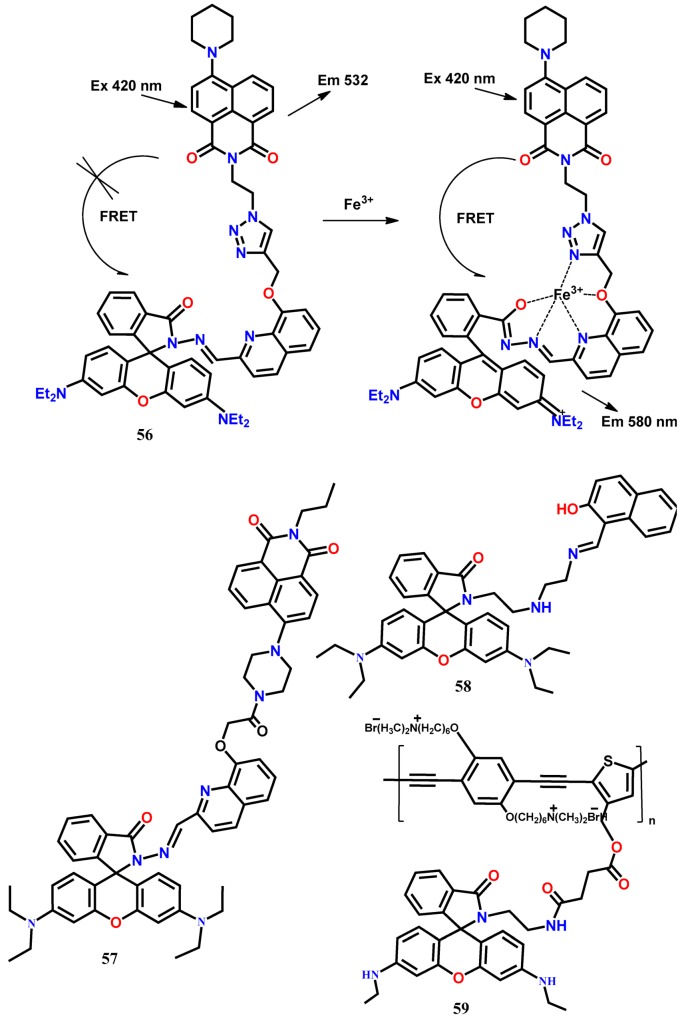





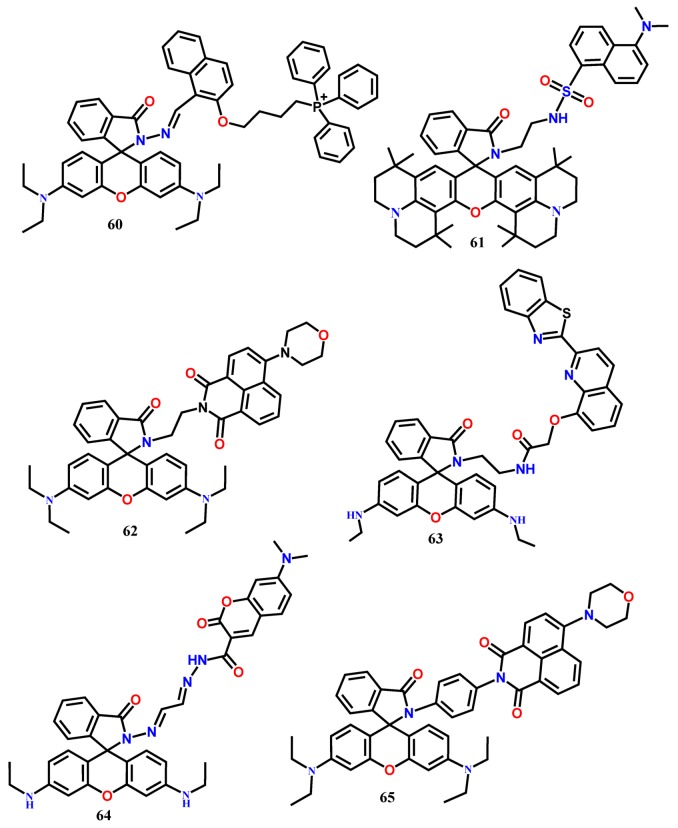



The FRET-based energy transfer mechanism is most popular for the designing of ratiometric fluorescent probes, and their FRET efficiency is primarily controlled by the spectral overlap between the emission spectrum of the energy donor and the absorption spectrum of the energy acceptor. In contrast to the FRET systems, the through-bond energy transfer (TBET) systems are not limited to such spectral overlap for the energy transfer between two fluorophores. The probes based on TBET mechanisms are well-known to show fast energy transfer rates and large pseudo-Stokes shift. In probe **65**, the energy donor (4-morpholine)-1,8-naphthalide moiety is linked to the energy acceptor rhodamine by a rigid and conjugated spacer *p*-phenylenediamine [73]. This rigid connection efficiently prevents the fluorescence quenching of naphthalimide. In the absence of Fe^3+^, the excited energy of the naphthalimide donor is not transferred to the closed form of rhodamine acceptor, and the characteristic peak of naphthalimide is observed at 535 nm in CH_3_OH-H_2_O (4:6, *v*/*v*). The complexation of **65** with Fe^3+^ opens the spirocyclic ring of the rhodamine ring, resulting in a significant fluorescence enhancement at 585 nm (λ_exc_ = 420 nm). Simultaneously, the naphthalimide emission at 535 nm quenches due to the TBET. The ratiometric probe **65** shows a linear fluorescence response from 0 to 20 μM with the LOD of 0.105 μM Fe^3+^. Further, the probe has been applied for ratiometric fluorescence imaging of Fe^3+^ ions in living EC109 cells (Figure 7).

Chattopadhyay and his co-workers [74] introduced the ratiometric fluorescent probe **66**, which undergoes a 1,5-sigmatropic shift in solution to form the benzimidazole derivative with the more chelating environment. The intensity of the weakly fluorescence benzimidazole derivative of **66** at 412 nm (λ_exc_ = 365 nm) was decreased, and a new fluorescence peak appeared at 445 nm upon addition of Fe^3+^ in CH_3_CN-HEPES buffer (1/4, *v*/*v*, pH 7.4) due to the chelation enhanced fluorescence (CHEF) effect. Similar ratiometric fluorescence changes were also observed in the presence of Fe^2+^. With **66**, the ratiometric fluorescence response could be used to detect Fe^3+^ and Fe^2+^ ions down to 3.5 µM and 2 µM, respectively. Further, the probe was applied to detect intracellular Fe^3+^ ions in live HeLa cells. Based on the efficient ligand metal charge-transfer effect, the piperazine-based dipodal fluorescent probe **67** appended with 8-hydroxyquinoline has been introduced for the ratiometric detection of Fe^3+^ [75]. In CHCl_3_-MeOH (1:1, *v*/*v*), the probe has shown monomer emission of the quinoline moiety at 400 nm (λ_exc_ = 300 nm). Upon addition of Fe^3+^, the emission peak at 400 nm is quenched, and a new broad emission appears at 480 nm. This ratiometric fluorescence quenching in the presence of Fe^3+^ is attributed to the strong interaction of Fe^3+^ with the triazolmethyloxyquinoline (as tridentate ligand) motifs of **67**. Calibrating the intensity ratio (I_480_/I_400_) with the change in concentration of Fe^3+^ gives LOD of 1.17 μM. Receptor **67** is cells permeable and detect intracellular Fe^3+^ ions in live HeLa cells with no cytotoxicity. Sequentially, the **67**-Fe^3+^ complex ensemble is applied for the selective detection of fluoride anion.

The dipodal clip-type ratiometric fluorescent probe **68** containing two benzimidazole groups has been applied for the selective detection of Cr^3+^ and Fe^3+^ ions in DMSO/H_2_O (1:99, *v*/*v*) [76]. When excited at 320 nm, the probe **68** fluorescence at 443 nm quenches and simultaneously enhances at 378/380 nm upon addition of Cr^3+^/Fe^3+^. It has been proposed that the blue-shift in the fluorescence of **68** is due to the intramolecular charge transfer (ICT), whereas the enhancement of fluorescence intensity upon Cr^3+^/Fe^3+^ complexation is most likely due to the inhibition of PET processes. With the probe, the minimum detection limit is estimated as 25 μM and 2 μM for Cr^3+^ and Fe^3+^, respectively. Further, the selective UV-Vis spectral changes of **68** in the presence of Fe^3+^ allow discriminating the presence of both Cr^3+^ and Fe^3+^. Using the robust fluorophore 2-pyridylthiazole and ICT mechanism, an easy-to-prepare ratiometric fluorescence probe **69** has been developed for the selective detection of Fe^3+^ [77]. In an aqueous system (CH_3_CN/Tris buffer = 9:1, *v*/*v*, pH = 7.4), the probe emission at 431 nm quenches with the concomitant appearance of a new emission at 517 nm (λ_exc_ = 380 nm) in the presence of Fe^3+^. The LOD of this probe is estimated to be 4.47 μM for Fe^3+^. Recently, the easy-to-prepare linear Schiff base receptor **70** has been developed for the ratiometric detection of Fe^3+^ and fluorescence turn-off sensing of Cu^2+^ in aqueous acetonitrile medium [78]. Upon addition of Fe^3+^, receptor **70** induces a selective fluorescence enhancement with a 22 nm red-shift from 504 nm to 526 nm, making it easily distinguishable from the other tested metal ions. The fluorescence enhancement is observed presumably due to deprotonation of the phenolic-OH protons on coordination with Fe^3+^, inhibiting the –C=N isomerization and/or the ESIPT process in the excited state. Also, the red-shift indicates that the possible ICT occurs in **70** on interaction with Fe^3+^. In contrast, the fluorescence of **70** quenches upon addition of Cu^2+^. The quenching by Cu^2+^ is most likely due to an energy transfer process occurring between **70** and paramagnetic Cu^2+^. From the emission titrations, the LOD for the sensing of Fe^3+^ and Cu^2+^ ions are estimated to be 10 nM and 15 nM, respectively. Besides, the organic nanoparticles (ONPs) of the probe **70** has been developed and applied for the detection of Fe^3+^ and Cu^2+^ in different drug supplements available in the market. Also, the probe fluorescence response has been applied to mimic the IMP (IMPLICATION) type logic gate with the two-inputs as Fe^3+^ and Cu^2+^.



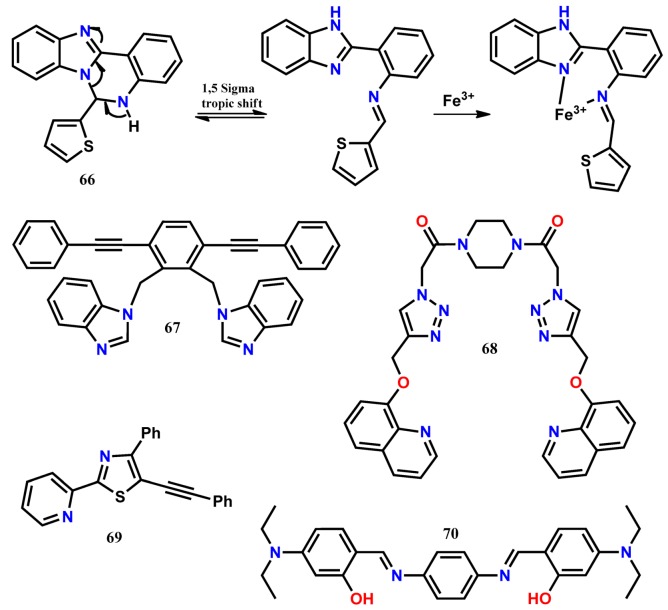



## 4. Fluorescent Chemodosimeters for Fe(III)

The design of Fe^3+^-selective fluorescent turn-on probes with high selectivity and sensitivity can be achieved by chemodosimeter approach, where the Fe^3+^ ions mediate the breaking of some important bonds in the probe, leading to the irreversible transduction of a detectable fluorescent signal [79]. Recently, a few one-time use fluorescent chemodosimeters are reported for the detection of Fe^3+^ (Table 4) and are also applied successfully to detect the intracellular Fe^3+^ ions in live cells by bioimaging.

Chen et. al. [80] introduced a novel chemodosimeter-based fluorescent probe **71**, consisted of a BODIPY dye, as a signal transducer that is suitably linked to a hydroxylamine unit (Figure 8). Above pH = 5.8, the electron-donating ability from the hydroxylamine to the fluorophore unit quenched the fluorescence at 615 nm (λ_exc_ = 585 nm) due to the PET. In HEPES aqueous buffer (pH 7, 40 mM), the addition of Fe^3+^ selectively oxidized the hydroxylamine that inhibited the PET process, and a significant fluorescence enhancement was observed (from *Φ* = 0.01 to *Φ* = 0.35). The probe **71** showed a good linear dependence of fluorescence intensity on Fe^3+^ concentration (0–50 μM) and applied successfully for the monitoring of Fe^3+^ concentration in live MCF-7 cells by using the confocal fluorescence microscope (Figure 8).

Subsequently, considering the ability of Fe^3+^ to mediate the deprotection of acetal reaction, the ratiometric fluorescent probe **72** has been designed for the highly selective detection of Fe^3+^ in 20 mM potassium phosphate buffer/acetone (pH 7, 1:4 (*v*/*v*)) at room temperature [81]. The acetal group of **72** is deprotected into aldehyde by Fe^3+^, increasing the π-electrons conjugation, and, therefore, the effective ICT from the phenanthroline unit to the aldehyde results in the red-shift in the emission band from 390 nm to 522 nm (*λ*_exc_ = 360 nm). The fluorescence titration of **72** with Fe^3+^ shows satisfactory linearity in the range of 0–30 μM between the emission ratio (*I*_522_/*I*_390_) and the concentration of Fe^3+^. With **72**, the concentration of Fe^3+^ can be detected down to 0.12 μM. Further, the probe has been successfully applied for imaging Fe^3+^ in living pancreatic cancer cells (Figure 9).

Sahoo and his co-workers [82] introduced an easy-to-prepare chemodosimeter-type optical chemosensor 73 for the selective detection of Fe^3+^ by condensing 1-aminopyrene with pyridoxal. Probe 73 showed weak emission at 441 nm (λ_exc_ = 325 nm) due to the PET process occurring from the pyridoxal imine to the pyrene fluorophore. Addition of Fe^3+^ hydrolyzed the imine linkage of 73, leading to the back-formation of 1-aminopyrene and pyridoxal, resulting in a significant fluorescence enhancement at 441 nm in the aqueous DMSO medium. The probe 73 showed good linearity from 0 M to 6.98 × 10^−5^ M with the limit of detection down to 4.3 μM for Fe^3+^. The probe 73 was highly specific for the detection of Fe^3+^, and the concentration of Fe^3+^ could be monitored by using both spectrophotometer and smartphone. Besides, the probe **73** could be applied to monitor Fe^3+^ within the live HeLa cells. With a similar approach, three more Fe^3+^-selective fluorescent probes **74**-**76** have been reported [83,84,85]. The probe **74** shows significant fluorescence enhancement at 440 nm (λ_exc_ = 396 nm) selectively in the presence of Fe^3+^ in DMSO/H_2_O (*v*/*v* = 70:30) due to the Fe^3+^-mediated hydrolytic cleavage of the imine linkage [83]. Using **74**, the concentration of Fe^3+^ can be detected down to 1.37 μM and successfully applied to detect intracellular Fe^3+^ ions in live RAW264.7 cells by imaging experiment. The same group reports the Fe^3+^-selective chemodosimeter **75**, showing selective fluorescence enhancement at 440 nm (λ_exc_ = 390 nm) in DMSO/H_2_O (*v*/*v* = 9/1, buffered with HEPES, pH = 7.4) [84]. The probe **75** shows nanomolar detection limit of 75.7 nM for Fe^3+^. The probe **75** shows good cell-membrane permeability, and also the Fe^3+^-directed hydrolysis of imine linkage is shown to be detected in live HeLa cells by fluorescence microscopy. Recently [85], the probe **76** is introduced for the selective detection of Fe^3+^ in MeOH/H_2_O (9/1, *v*/*v*). The weakly emissive probe **76** shows gradual fluorescence enhancement at 430 nm upon successive incremental addition of Fe^3+^ due to the back-formation of 2,5-dimethoxybenzaldehyde and 1-aminopyrene. The probe **76** shows the LOD of 0.118 μM without any interference from other tested metal ions. Similar to the other chemodosimeters, probe **76** has been successfully applied to detect intracellular Fe^3+^ ions in live RAW264.7 cells by fluorescence microscopy. Adopting the hydrolytic cleavage of imine linkage, the multi-analytes selective chemodosimeter **77** has been developed for the detection of trivalent metal ions (Fe^3+^, Cr^3+^, and Al^3+^) in THF-H_2_O (8:2) medium [86]. The strong Lewis acidity of trivalent cations (Fe^3+^/Al^3+^/Cr^3+^) selectively breaks the imine linkage, resulting in significant fluorescence enhancement at 430 nm (λ_exc_ = 330 nm). With the probe, the concentration of selective cations Fe^3+^, Al^3+^, and Cr^3+^ can be detected down to 0.38 nM, 0.38 nM, and 0.36 nM, respectively. Further, the probe has been applied to image the native cellular iron pools in *Candida albicans* cells.



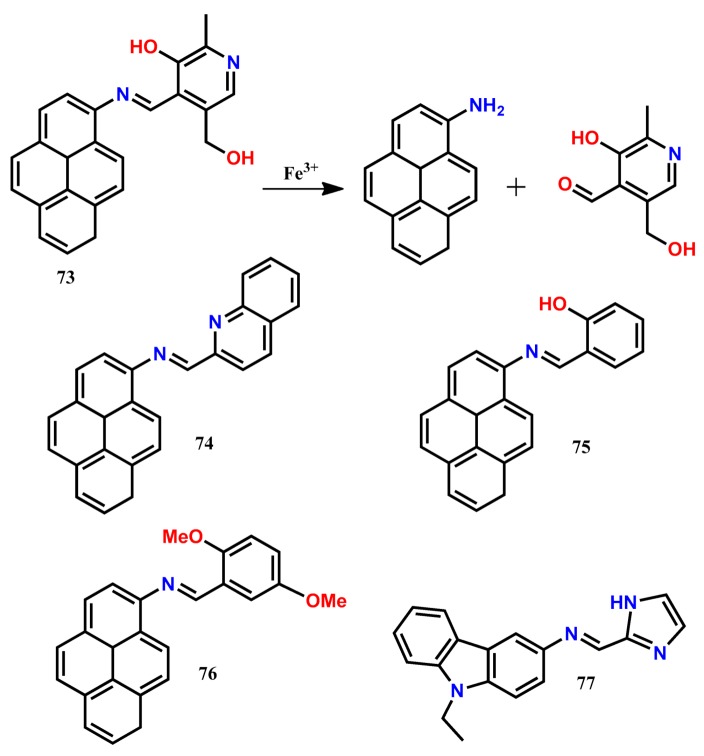



## 5. Conclusions

Because of the quenching effects of Fe^3+^, the designing of fluorescent turn-on and ratiometric probes are very challenging, and, therefore, we observed during the literature search that the majority of the reported fluorescent probes for Fe^3+^ are based on fluorescence quenching process. However, by applying the sensing mechanisms like the complexation-induced opening of the spirocyclic ring, PET, FRET, TBET, AIEE, excimer formation, etc., several Fe^3+^-selective fluorescent turn-on and ratiometric probes have been reported after 2012. This critical review presents a total of **77** Fe^3+^-selective fluorescent turn-on and ratiometric probes that can monitor Fe^3+^ ions, both in solution as well as within live cells. The analytical parameters like selectivity, specificity, and sensitivity of the summarized probes are suitable for their potential applications in monitoring Fe^3+^ ions concentration in various real environmental and biological samples. We believe the advantages of fluorescent probes like low-cost and simplicity would encourage the real applications of the probes summarized in this review. However, despite several analytical advantages, further research is required to develop probes that function in a pure aqueous medium because the use of organic solvents can limit their use in biological samples. Therefore, there is a wide-open scope for further research to develop novel Fe^3+^-selective fluorescent probes with potential analytical applications.

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
