# Peer review of "Recent Advances on Iron(III) Selective Fluorescent Probes with Possible Applications in Bioimaging"

_molecules, 2019, doi:10.3390/molecules24183267_

Round 1

Reviewer 1 Report

Authors prepared a review paper regarding recently developed fluorescent probes for the detection of iron (III) ion and their applications. Recently reported iron ion sensing probes (77 probes) are summarized in this manuscript with photophysical property of each probes and fluorescence imaging results. However, I think this manuscript should be revised, because I can’t find any useful information in this paper. In addition, there are many typos and grammar errors. The figures and tables also need to be re-drawn and re-categorized. I would like to recommend the major revision of this manuscript with several reasons as below.

There is many typos and grammar errors. The tables in this manuscript should be re-organized for better understanding. The Figures from original article should be cited with proper explanation – not “adapted from ref. xx”. The permission issue is very important in the review paper. The introduction part and conclusions are very weak. In the introduction part, author have to mention the the difference of iron ion probe review papers which are already published, and emphasize what is important points of this manuscript. In addition, the perspective should be described in the conclusion part. At this stage, there is no new information in terms of “Recent Advances” for iron(III) sensing probe. General binding mode of Schiff-based fluorescent probe should be described as a scheme. The sensing mechanism have to be categorized – (i) binding-based, (ii) reaction-based, (iii) etc? Minor points

- page 3, line 59: should be one paragraph

- page 4, line 107 (Table 1): detection limit should be mentioned

- page 8, line 133, 134: the concentration should be “nM”

- page 9, line 148: the unit of detection limit should be unified. (M, nM, ppb, ppm, etc)

- page 9, line 155: the reason why probe 15 does not show fluorescence above pH 6, because it is not possible to use in biological system.

- page 25, line 586: probe 77 works in organic solvent (80% THF) and have no selectivity toward iron(III) ion. What are the merits of this probe? It should not be worked in biological system.

- Probe 1~40 are summarized in Table 1. Author need to prepare Table 2 for the probe 41 ~ 77.

Author Response

We are thankful to the reviewer for the valuable suggestions to improve our review article. The manuscript was edited carefully and the grammatical errors were corrected. Necessary permission to re-use the figures from other publishers were taken and cited with proper explanation. The introduction and conclusion sections were revised as per the reviewer suggestion, and the importance of the review was presented. For simplicity, the probes were summarized in three main sections: (i) turn-on, (ii) ratiometric and (iii) chemodosimeter (reaction-based turn-on). Other suggested page wise errors were corrected as mentioned below:

page 3, line 59: should be one paragraph…Done

Page 4, line 107 (Table 1): detection limit should be mentioned…Done ..Table 1 was modified

Page 8, line 133, 134: the concentration should be “nM”…Done

Page 9, line 148: the unit of detection limit should be unified. (M, nM, ppb, ppm, etc)…Done in page 6 and also corrected in whore revised review.

Page 9, line 155: the reason why probe 15 does not show fluorescence above pH 6, because it is not possible to use in biological system ….The sentence was modified as ‘the probe 15 alone does not show fluorescence above pH 6

Page 25, line 586: probe 77 works in organic solvent (80% THF) and have no selectivity toward iron(III) ion. What are the merits of this probe? It should not be worked in biological system…Well this probe was added due to the application of the probe for the detection of Fe3+ within live cells.

Probe 1~40 are summarized in Table 1. Author need to prepare Table 2 for the probe 41 ~ 77…Done.

Reviewer 2 Report

Molecule

This review paper is talking about “iron(III) selective fluorescent probes”. It is better to discuss the importance of iron(III) in the introduction. Also, the general design strategy of iron(III) selective fluorescent probes is very important. It is nice to have a discussion about the general design strategy of iron(III) probe in the introduction.

Some errors are found in the manuscript.

In Scheme 1. (B), The “N” is missing in the complex. In Figure 2, each figure is labeled as “a, b, c, d, e, f”, but in the caption, “A, B, C, D, E, F” are used for each figure. In the structure of compound 41, the “N” in BODIPY is missing. In the structure of compound 56, the C=N bond in Fe(III) complex is missing. In the structure of compound 67, two “H” is unnecessary to be drawn. In the structure of compound 71, the C=C bond in BODIPY is missing.

Author Response

Thank you for the valuable suggestions to improve our review. According to your suggestions, we have edited the introduction section and the importance of the aim of this review was narrated. Appropriate reference was cited for the various mechanism adopted for the design of fluorescent sensors.

Also the errors pointed in the structures are corrected.

Round 2

Reviewer 1 Report

I think all the issues are addressed in this version. I would like to recommend the acceptance of this manuscript.